# Nutrient availability is a dominant predictor of soil bacterial and fungal community composition after nitrogen addition in subtropical acidic forests

**Juyan Cui**[1,2,3], **Xiaochun Yuan**[1,2,4], **Qiufang Zhang**[1,2], **Jiacong Zhou**[1,2], **Kaimiao Lin**[4], **Jianguo Xu**[5], **Yaozhong Zeng**[1,2], **Yue Wu**[1,2], **Lei Cheng**[1,2], **Quanxin Zeng**[1,2], **Kongcan Mei**[1,2], **Yuehmin Chen**[1,2,6]*

1 College of Geographical Science, Fujian Normal University, Fuzhou, China, 2 State Key Laboratory of Subtropical Mountain Ecology, Fujian Normal University, Fuzhou, China, 3 Architectural Engineering Institute, Tongling University, Tongling, China, 4 Tourism College, Wuyi University, Wuyishan, China, 5 Daiyun Mountain National Nature Reserve Administration Bureau, Quanzhou, China, 6 Institute of Geography, Fujian Normal University, Fuzhou, China

* ymchen@fjnu.edu.cn

**Data Availability Statement:** Data are available from the NCBI Sequence Read Archive (accession number PRJNA666625).

**Funding:** The research was supported by the National Natural Science Foundation of China (No.

## Abstract

Nutrient addition to forest ecosystems significantly influences belowground microbial diversity, community structure, and ecosystem functioning. Nitrogen (N) addition in forests is common in China, especially in the southeast region. However, the influence of N addition on belowground soil microbial community diversity in subtropical forests remains unclear. In May 2018, we randomly selected 12 experimental plots in a *Pinus taiwanensis* forest within the Daiyun Mountain Nature Reserve, Fujian Province, China, and subjected them to N addition treatments for one year. We investigated the responses of the soil microbial communities and identified the major elements that influenced microbial community composition in the experimental plots. The present study included three N treatments, i.e., the control (CT), low N addition (LN, 40 kg N ha$^{-1}$ yr$^{-1}$), and high N addition (HN, 80 kg N ha$^{-1}$ yr$^{-1}$), and two depths, 0−10 cm (topsoil) and 10−20 cm (subsoil), which were all sampled in the growing season (May) of 2019. Soil microbial diversity and community composition in the topsoil and subsoil were investigated using high-throughput sequencing of bacterial 16S rDNA genes and fungal internal transcribed spacer sequences. According to our results, 1) soil dissolved organic carbon (DOC) significantly decreased after HN addition, and available nitrogen (AN) significantly declined after LN addition, 2) bacterial α-diversity in the subsoil significantly decreased with HN addition, which was affected significantly by the interaction between N addition and soil layer, and 3) soil DOC, rather than pH, was the dominant environmental factor influencing soil bacterial community composition, while AN and MBN were the best predictors of soil fungal community structure dynamics. Moreover, N addition influence both diversity and community composition of soil bacteria more than those of fungi in the subtropical forests. The results of the present study provide further evidence to support shifts in soil microbial community structure in acidic subtropical forests in response to increasing N deposition.

31670620) and Natural Science Foundation of Fujian Province(No. 2019J05163).

**Competing interests:** The authors have declared that no competing interests exist.

## Introduction

Industrialization and urbanization have increased the quantity of nitrogen (N) and phosphorus entering terrestrial systems [1], especially in China's warm and humid climatic zones [2]. The annual atmospheric N deposition in China increased from 7.6 to 20 Tg in 1978–2010 [3], attracting the attention of researchers regard the potential effects of N deposition on terrestrial ecosystems. Excessive N input negatively impacts the structure and functioning of terrestrial ecosystems, causing soil acidification [4, 5], decline in tree productivity [6–8], reduced plant diversity [9, 10], and reduced understory vegetation species richness [11, 12]. N deposition could also alter ectomycorrhizal fungal [13] and soil bacterial community structure [14, 15].

Soil bacteria and fungi play pivotal roles in terrestrial ecosystems, mediating biogeochemical processes (e.g., carbon [C] and N cycling) [14, 16] and promoting aboveground plant health and productivity [2, 17, 18]. Several meta-analyses have examined the effects of N addition on soil microbial biomass and community composition [19–28]. Overall, simulated N deposition appears to suppress microbial biomass [2]; however, the effects varied across biomes. N addition reduced soil microbial biomass C (MBC) in temperate forests and grasslands, but significantly increased MBC in tropical/subtropical forests [27]. This increase in microbial biomass following N addition is thought to be mediated by increasing C or N resource availability [28].

N addition can also induce changes in soil microbial α-diversity and microbial community composition [29–31]. In a previous meta-analysis, N additions increased the Shannon indices and reduced bacterial Chao1 indices, although the effect on soil bacterial richness was greater than that on fungal richness [28]. Similarly, in a subtropical forest, simulated N deposition significantly decreased microbial α-diversity [29, 30, 32]. However, N additions do not always alter in the evenness and richness of soil bacterial and fungal communities. Short-term N addition significantly altered soil microbial community structure by increasing fungi/bacteria ratio (F/B) in tropical/subtropical forest soil, although long-term N addition did not induce such changes [29, 33]. Nevertheless, some studies have reported no change in the F/B ratio in response to short-term N enrichment [34], while others have demonstrated that N enrichment could reduce the F/B ratio in subtropical forests [28, 35]. According to the copiotrophic hypothesis, increasing N enrichment could decrease the abundance of oligotrophic groups but increase the abundance of copiotrophic groups [14, 15, 32, 36–38]. Therefore, the effects of N enrichment on microbial communities are variable, and the mechanisms by which N addition influence the soil microbial community composition needs to be elucidated.

N addition can influence soil physicochemical properties, which, in turn, influence soil microbial structure [39]. Numerous studies have explored the major factors that could explain the changes observed in sensitive soil microbial communities within terrestrial ecosystems following N enrichment [5]. For example, soil pH is considered a key factor influencing soil bacterial community composition in terrestrial ecosystems [38, 40]. However, fungal community composition appears to be less impacted by pH, because fungi usually exhibit a greater range of optimal pH than that in bacteria [40, 41].

The absolute abundances of major microbial groups present in the soil is positively correlated with soil C and N concentrations [42]. N addition could directly influence soil bacterial community composition by altering the availability of compounds such as modified ammonium N ($NH_4^+$-N) [5, 28, 43]. Furthermore, soil bacterial community composition could be influenced by a slight decline in pH caused by an increase in soil $NH_4^+$-N in extremely acidic soil (pH < 4.5) [5]. Although such mechanisms have been previously studied, few comprehensive studies have examined the wide array of soil physicochemical properties associated with

changes in bacterial and fungal communities following N enrichment in subtropical forest ecosystems.

Currently, subtropical forests in Southeast China are experiencing extensive N deposition, accompanied by signs of N saturation, leading to soil acidification [44]. In the present study, we investigated the responses of bacterial and fungal communities to N addition and the underlying mechanisms of such responses. We addressed the following questions in this study: (1) How do bacterial and fungal communities respond to N additions? (2) How do biotic and abiotic elements in the soil modulate the responses of bacterial and fungal communities? (3) What are the potential mechanisms that are responsible for the observed changes in microbial community structure? We hypothesized that N addition could reduce the bacterial and fungal diversity, and, thereby alter soil microbial community composition in subtropical forests. We also hypothesized that soil pH and nutrients would regulate the responses of microbial communities through the direct or indirect effects of N amendments.

## Materials and methods

### Site description and experimental design

The study area is located in the scenic Jiuxian Mountains within the Daiyun Mountain Nature Reserve in Dehua County, Fujian Province, in southeast China (118˚06′3–5″E, 25˚42′22–27″N). The reserve is located in the transition zone between central subtropical and southern subtropical forests [45]. This nature reserve, which has the southernmost distribution in China, has the highest degree of biodiversity per unit area in China, and comprises the largest area of the best preserved natural *Pinus taiwanensis* community (Fig 1). *P. taiwanensis* is a unique alpine tree species in China that plays a valuable role in ecological restoration. The area is also the largest germ plasm gene pool of *P. taiwanensis* in China, which are mainly distributed at an altitude of 1000−1800 m. The climate type is subtropical maritime monsoon climate, with cold winters and hot summers. The climate can be simultaneously hot and rainy, with four distinct seasons, mean annual temperature of 20˚C, a mean annual precipitation of 1800 mm, and a mean relative humidity of 80%. The soil is an Ultisol formed from sandstone and classified as red soil according to Chinese soil classification. Total N deposition in the region is approximately 38 kg N ha$^{-1}$ yr$^{-1}$ [44].

In the *P. taiwanensis* forest, we randomly selected 12 experimental plots to perform the simulated N addition treatment in May 2018. No permits were required for the experiments since this study was part of an on-going collaborative scientific effort with the Daiyunshan Nature Reserve. All experimental plots are close to the top of the mountain, face the same direction, and have similar slopes and elevations. In each experimental plot, 12 subplots (10 × 10 m) were randomly assigned three different levels of N addition, including the control (CT), low N (LN), and high N (HN) (0, 40, and 80 kg N ha$^{-1}$ yr$^{-1}$, respectively), with four replicates for each N level. Each plot was surrounded by a 5-m wide buffer zone. We used urea [$CO(NH_2)_2$] as the N source, which was added into the experimental plots from March to September every year. The required amounts of urea were dissolved in 8 L purified water, and CT plots received an equivalent volume of water without urea.

### Soil sampling and analysis

In May 2019, soil samples were randomly collected from 5−8 cores at depths of 0−10 cm (topsoil) and 10−20 cm (subsoil) from each plot after removing the surface litterfall. Soil samples were passed through a 2-mm sieve after removing the litter, roots, and stones, and divided into three parts. One portion of each sample was immediately processed to measure soil moisture content (SMC), mineral N, and dissolved organic C (DOC), and then stored at -80˚C for the

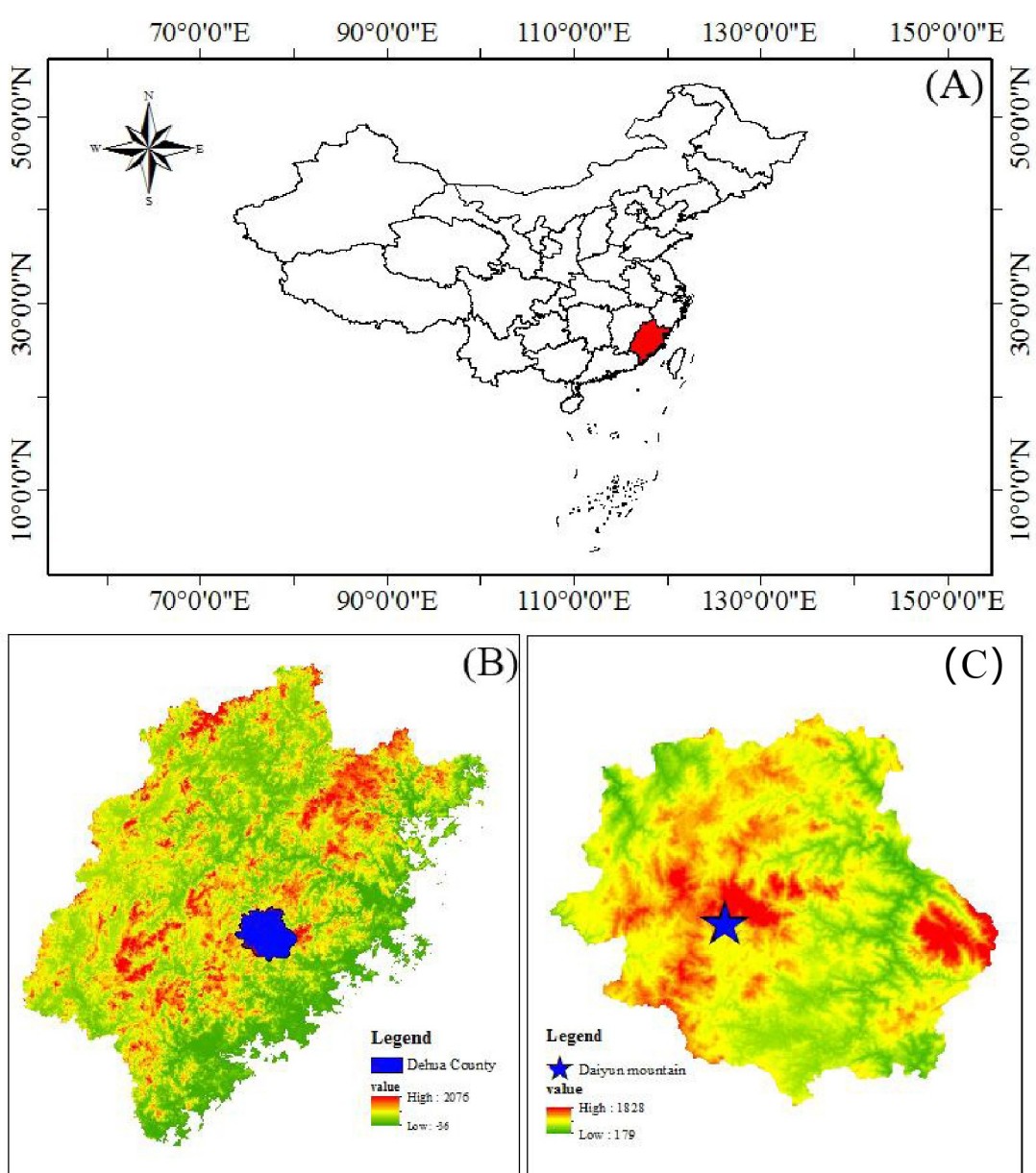

**Fig 1. Study site location.** (A) Map of China. (B) Map of Fujian Province. (C) Map of Dehua county. The blue pentagram is the sampling area.

soil molecular biology analyses. Another portion was air-dried for soil physicochemical parameter analyses. The air-dried soil portion was passed through a 0.149-mm sieve to measure total soil C and N.

Soil pH was determined using a glass electrode (STARTER 300; OHAUS, USA). Samples were shaken for 0.5 h in a 1:2.5 soil:water solution (w/v). Soil moisture content was assessed following oven-drying for 48 h at 105°C to constant mass. The soil organic C (SOC) to total N ratio (C:N) was measured using an elemental analyzer (Elementar Vario EL III; Elementar, Langenselbod, Germany). Total P content was determined after digesting the samples with $H_2SO_4$ and $HClO_4$ (at a 4:1 ratio) using a continuous flow analyzer (Skalar san++, Skalar, Breda, Netherlands). After 2 M KCl extracts of fresh soil samples were prepared, soil

suspensions were centrifuged at 4000 rpm for 10 min, filtered through Whatman 42 filter papers, and analyzed for $NH_4^+$-N and $NO_3^-$N using the Continuous Flow Analytic System (Skalar san++; Skalar, Breda, the Netherlands). Available N (AN) was calculated as the sum of ammonium-N and nitrate-N. DOC was extracted with deionized water in a 1:4 soil:water solution (w/v) by shaking for 0.5 h, centrifuging at 4000 rpm for 0.5 h, and then filtering through 0.45-μm Millipore filters. The extracts were analyzed on a TOC-VCPH/CPN analyzer (Elementar Analysensysteme GmbH, Germany) to determine the DOC concentrations. The soil cation exchange capacity (CEC) was measured using the ammonium acetate method [46].

Soil MBC and N (MBN) were measured using the chloroform-fumigation extraction method [47]. The correction factor of MBC was 0.38 [48], while that of MBN was 0.45 [49].

## Quantitative PCR (qPCR) analysis of soil microbial abundance

Soil DNA was extracted from 0.25-g soil samples using PowerSoil DNA Isolation Kit (MoBio Laboratories, Carlsbad, CA) according to the manufacturers instruction. Genomic DNA purity and quality were checked on 0.8% agarose gels. The quantitative PCR (qPCR) method was used to measure the relative abundance of the bacterial 16S rDNA genes and fungal internal transcribed spacers (ITS) that were amplified using the primers detailed in S1 Table [50].

For each soil sample, a 10-digit barcode sequence was added to the 5´ end of the forward and reverse primers (provided by Allwegene Company, Beijing). PCR was carried out on a Mastercycler Gradient (Eppendorf, Germany) using 25-μl reaction volumes, containing 12.5 μl 2× Taq PCR MasterMix, 3 μl bovine serum albumin (2 ng μl$^{-1}$), 2-μl primers (5 μM), 2-μl template DNA, and 5.5-μl ddH$_2$O. The cycling parameters were as follows: 94˚C for 5 min, followed by 30 cycles of 94˚C for 30 s, 55˚C for 30 s, and 72˚C for 30 s, with a final extension at 72˚C for 10 min. Three PCR products per sample were pooled to mitigate reaction-level PCR biases. PCR products were purified using a QIAquick Gel Extraction Kit (QIAGEN, Germany) and quantified using real time PCR. The fungal-to-bacterial (F/B) ratio was calculated using the 16S rDNA and ITS gene copy numbers.

## Illumina Miseq sequencing and bioinformatics analysis

Deep sequencing was performed on a Miseq platform at Allwegene Company (Beijing). After the run, image analysis, base calling, and error estimation were performed using Illumina Analysis Pipeline v2.6 (Illumina, San Diego, CA, US).

Raw data were first split based on barcodes and then screened. Sequences were removed from consideration if they were shorter than 200 bp, had a low-quality score per sequence ($\leq$ 20), contained ambiguous bases, or did not exactly match the primer sequences and barcode tags. The data were split then overlapped and spliced using PEAR (Paired-End read merger) software. Qualified reads were separated using the sample-specific barcode sequences and trimmed with Illumina Analysis Pipeline Version 2.6, while chimeras were removed. Subsequently, the dataset was analyzed using QIIME v1.8.0 (http://qiime.org). Sequences were clustered into operational taxonomic units (OTUs) at a similarity level of 97% [51] to generate rarefaction curves and to calculate the richness and diversity indices. The ribosomal database project classifier tool (release 10.3) was used to classify all sequences into different taxonomic groups [52].

## Statistical analysis

All data are presented as means in the tables and figures. Statistical analyses were performed using IBM SPSS Statistics 20 (IBM Corp., Armonk, NY, US), and data were tested for normality and homoscedasticity before statistical analyses. One-way analysis of variance (ANOVA) was used to determine the differences in soil properties and soil microbial parameters, and

multiple comparisons were conducted using the least significant difference (LSD) test at $p < 0.05$. Moreover, correlation analysis was performed with the Pearson's test (two-tailed) at two significance levels, i.e., $p < 0.05$ and $p < 0.01$.

To examine similarity between different samples, we used clustering and Principal Components Analysis (PCA) on the OTUs from each sample in R [53]. We used PCA because it is suitable for ordination using only species composition, and is an unconstrained method. The evolutionary distances between microbial communities from each sample were calculated using the Tayc coefficient and represented as an unweighted pair group method with arithmetic mean (UPGMA) clustering tree describing the dissimilarity (1—similarity) between multiple samples [54]. Multivariate statistical analysis, known as linear discriminant analysis (LDA), was performed to calculate the effect sizes in order to identify the species with significant differences in abundance among the soil samples.

Two-way ANOVA was used to compare the results in the soil fractions (topsoil and subsoil) and fertilizer treatments. Pearson's correlation analysis was performed to assess the relationships between soil properties and microbial α-diversity indices. Chao1, Simpson, Shannon, phylogenetic diversity whole tree (PD whole tree), and observed species indices were calculated using QIIME v1.8.0 to estimate the α-diversity levels. Partial least squares discriminant analysis (PLS-DA), which is a multivariate statistical analysis method for discriminant analysis, estimated the β-diversity in soil bacterial and fungal communities. Pearson's correlation analysis was used to address the relationships among soil properties, microbial abundance, and microbial biomass. In all tests, a $p < 0.05$ was considered statistically significant. To identify the relationship between changes in the soil microbial community composition and soil environmental factors, redundancy analysis (RDA) and Mantel test analysis were performed using CANOCO 5.0 (Ithaca, NY, USA) and R (3.6.2), respectively.

## Results

### Responses of soil properties and microbial biomass to nitrogen additions

Among the soil physicochemical properties we examined, soil pH, SMC, CEC, SOC, total N (TN), TP, and C:N were not affected by N additions (Table 1). Meanwhile, soil $NO_3^-N$

**Table 1. Responses of physicochemical properties of soil at different depths to nitrogen addition in the Daiyun Mountain Nature Reserve in southeastern China.**

|  | Topsoil (0–10 cm) | | | | Subsoil (10–20 cm) | | | |
|---|---|---|---|---|---|---|---|---|
|  | CT | LN | HN | p | CT | LN | HN | p |
| pH | 4.22(0.04) | 4.32(0.05) | 4.35(0.10) | 0.43 | 4.57(0.04) | 4.6(0.03) | 4.58(0.03) | 0.80 |
| SMC (%) | 74(9.29) | 66(3.20) | 59(3.60) | 0.31 | 41(2.53) | 39(4.02) | 39(3.77) | 0.92 |
| CEC (cmol kg$^{-1}$) | 12.58(2.27) | 9.47(1.83) | 10.63(0.97) | 0.49 | 12.27(1.65) | 8.61(0.69) | 8.79(0.90) | 0.09 |
| NH$_4^+$ (mg kg$^{-1}$) | 74.28(8.22)[a] | 51.23(2.91)[b] | 62.77(5.42)[ab] | 0.06 | 30.32(3.98)[a] | 16.19(3.55)[b] | 29.61(3.05)[a] | 0.02 |
| NO$_3^-$ (mg kg$^{-1}$) | 4.80(1.56)[b] | 5.69(1.60)[ab] | 9.65(0.89)[a] | 0.07 | 6.22(0.29) | 5.49(0.72) | 6.45(0.55) | 0.47 |
| AN (mg kg$^{-1}$) | 79.08(7.28)[a] | 56.92(7.79)[b] | 72.42(10.25)[ab] | 0.03 | 36.54(4.23)[a] | 21.68(2.97)[b] | 36.06(2.80)[a] | 0.04 |
| DOC (mg kg$^{-1}$) | 325.75(46.54)[a] | 245.37(11.86)[ab] | 206.87(18.36)[b] | 0.05 | 63.92(7.84)[a] | 46.07(4.38)[ab] | 28.51(4.56)[b] | 0.01 |
| SOC (g kg$^{-1}$) | 55.73(2.68) | 55.99(4.41) | 47.02(3.16) | 0.17 | 26.25(2.96) | 27.27(3.92) | 30.98(3.77) | 0.63 |
| TN (g kg$^{-1}$) | 4.06(0.47) | 3.75(0.36) | 3.12(0.19) | 0.22 | 1.81(0.21) | 1.62(0.24) | 2.20(0.27) | 0.27 |
| TP (g kg$^{-1}$) | 0.25(0.03) | 0.22(0.02) | 0.25(0.01) | 0.51 | 0.17(0.02) | 0.14(0.01) | 0.19(0.03) | 0.21 |
| C:N | 14.98(0.50) | 15.01(0.40) | 15.11(0.64) | 0.98 | 14.55(0.09) | 14.82(1.11) | 14.11(0.58) | 0.75 |
| MBC | 543.98(45.42) | 682.61(119.04) | 616.18(127.69) | 0.66 | 283.83(53.21) | 360.33(98.88) | 437.41(59.25) | 0.38 |
| MBN | 97.09(6.09)[c] | 201.99(8.58)[a] | 145.27(9.75)[b] | 0.00 | 50.31(13.23) | 61.78(9.67) | 81.91(7.73) | 0.15 |

The mean values of soil properties in control (CT), low (LN), and high (HN) nitrogen addition treatments in topsoil (0–10 cm) and subsoil (10–20 cm) soil samples are shown. Different letters represent significant differences (one-way ANOVA, $p < 0.05$, LSD post hoc analysis) between different levels of nitrogen addition.

concentration in the topsoil ranged from 4.8 mg kg$^{-1}$ in the CT plots to 9.65 mg kg$^{-1}$ in the HN plots, and increased significantly under high-N addition, while soil $NH_4^+$-N concentration in the topsoil ranged from 74.28 mg kg$^{-1}$ in the CT plots to 51.23 mg kg$^{-1}$ in the LN plots and decreased significantly under LN treatments. Furthermore, LN addition reduced soil AN concentration and HN addition reduced soil DOC concentration in both the topsoil and the subsoil ($p < 0.05$). However, changes in MBN in response to N addition were only observed in the topsoil, while N addition did not affect MBC in the topsoil or the subsoil.

Bacterial and fungal abundances were determined by quantification of copy numbers of bacterial 16S rDNA or fungal ITS using qPCR. The highest number of 16S rDNA gene copies in both the topsoil and the subsoil were observed in the LN plots, while the lowest numbers were observed in the topsoil and the subsoil of the HN plots (S2 Table). F/B values were greater in the topsoil than in the subsoil (S2 Table). Additionally, F/B values were positively correlated with soil MBN ($R^2 = 0.58$, $p < 0.01$) and NO3—N ($R^2 = 0.51$, $p < 0.05$) levels, and negatively correlated with soil pH values ($R^2 = -0.61$, $p < 0.01$) (S3 Table).

### Effects of N addition on bacterial and fungal diversity

Overall, 3,760 OTUs were obtained from 1,185,330 high quality and chimera-free clean tags by Miseq sequencing of 16S rDNA gene amplicons, with an average of 19,719−89,416 clean tags per sample. Similarly, 4,584 OTUs were obtained from a total of 1,171,127 high quality and chimera-free clean tags by Miseq ITS sequencing, with an average of 23,183−86,895 clean tags per sample. The rarefaction curves showed that these are reasonable amounts of sequencing data (S1A and S1B Fig) with a Good's coverage of 97.13−98.75%, which indicated that sequence reads were sufficient to capture the bacterial and fungal α-diversity (S4 Table).

Soil bacterial and fungal α-diversity was affected differently by N addition between the topsoil and the subsoil (Table 2; S5 Table). N addition decreased bacterial α-diversity (Chao1, observed species, PD whole tree, Shannon indices) in the subsoil of the HN plots relative to that of the CT plots (S5 Table; $p < 0.05$). However, bacterial richness indices, including Chao1, observed species, and PD whole tree, in samples of the topsoil of LN plots were all

**Table 2. Effects of nitrogen deposition, soil layer, and their interactions on soil bacterial (B) and fungal (F) α diversity analyzed by two-way ANOVA.**

| Diversity indices | Soil layer | | Nitrogen treatment | | Interaction | |
|---|---|---|---|---|---|---|
| | F | p | F | p | F | p |
| B_Chao1 | 2.98 | 0.10 | 2.69 | 0.09 | 1.22 | 0.32 |
| B_goods_coverage | 3.17 | 0.09 | 4.60 | 0.02* | 0.72 | 0.50 |
| B_observed_species | 2.02 | 0.17 | 1.66 | 0.22 | 3.54 | 0.05* |
| B_PD_whole_tree | 0.001 | 0.97 | 1.57 | 0.24 | 5.80 | 0.01** |
| B_Shannon | 9.37 | 0.007** | 0.70 | 0.51 | 5.67 | 0.01** |
| B_Simpson | 11.85 | 0.003** | 0.37 | 0.70 | 3.51 | 0.05* |
| F_Chao1 | 24.24 | 0.00*** | 1.39 | 0.27 | 1.65 | 0.22 |
| F_goods_coverage | 23.8 | 0.00*** | 3.58 | 0.04* | 1.14 | 0.34 |
| F_observed_species | 12.85 | 0.002** | 1.10 | 0.36 | 1.48 | 0.25 |
| F_PD_whole_tree | 13.14 | 0.002** | 1.10 | 0.35 | 1.81 | 0.19 |
| F_Shannon | 1.33 | 0.263 | 0.05 | 0.95 | 0.76 | 0.48 |
| F_Simpson | 1.31 | 0.267 | 0.35 | 0.71 | 0.67 | 0.52 |

*$p < 0.05$

**$p < 0.01$

***$p < 0.001$.

slightly higher than in those of the topsoil of CT and HN plots, though the differences were not statistically significant ($p > 0.05$) (S5 Table). The interaction between N addition and soil layer significantly affected α-diversity of bacteria (observed species, PD whole tree, Shannon, and Simpson) (Table 2). The fungal α-evenness (Shannon, Simpson) and α-richness indices (Chao1 and PD whole tree), were significantly higher in the topsoil than in the subsoil. However, the indices declined in the topsoil but increased in the subsoil with N addition (S5 Table).

Pearson's correlation coefficients indicated that soil microbial α-diversity was correlated with soil properties (Fig 2A & 2B). Changes in soil bacterial α-diversity (Shannon and Simpson index) induced by N addition were most closely related to soil pH ($R^2$ = -0.47, $p < 0.05$; $R^2$ = -0.49, $p < 0.05$) and $NO_3^-N$, with a strong positive correlation with $NO_3^-N$ ($R^2$ = 0.47, $p < 0.05$; $R^2$ = 0.40, $p < 0.05$) observed. Changes in soil bacterial α-richness (Chao1 index) exhibited negative and positive correlations with soil pH and SOC, respectively ($R^2$ = -0.44, $p < 0.05$; $R^2$ = 0.44, $p < 0.05$). However, it was weakly correlated with C:N, CEC, TP, and AN. The pH of the soil was negatively correlated with fungal Chao1, observed species, and PD whole tree indices, and positively correlated with the Good's coverage index ($p < 0.01$). Soil properties (SMC, $NH_4^+$, AN, SOC, TN, DOC) exhibited significant positive correlations with fungal diversity indices (Chao1, observed species, PD whole tree), while they exhibited a negative correlation with Good's coverage index ($p < 0.01$). However, the correlations between Shannon and Simpson indices, and soil properties, were weak. Moreover, PLS-DA, which estimates the distances among multiple samples (β-diversity), revealed that soil bacterial and fungal communities differed significantly among N addition treatments (S2A–S2D Fig).

## Relative abundance of dominant microbial taxa and species variation

The relative abundances of dominant bacterial and fungal phyla in the topsoil and subsoil are shown in Fig 3. The variation in bacterial and fungal species under different treatments is illustrated in S3 Fig in cladograms constructed based on LDA analysis.

In LN plots, the bacterial phyla of Proteobacteria, Acidobacteria, and Actinobacteria constituted 80.25% and 74.22% of the total sequences in the topsoil and the subsoil, respectively, followed by Firmicutes (3.47% and 2.2%, respectively), Planctomycetes (4.3% and 6.62%, respectively), and Chloroflexi (3.23% and 8.24%, respectively). In the HN plots, the bacterial phyla of Proteobacteria, Acidobacteria, and Actinobacteria constituted 76.95% and 77.25% of the total sequences in the topsoil and subsoil, respectively, followed by Firmicutes (4.89% and 0.74%, respectively), Planctomycetes (4.22% and 5.97%, respectively), and Chloroflexi (2.87% and 9.62%, respectively) (Fig 3). Among fungi, the phylum Ascomycota was the most abundant in the LN plots, constituting 46.9% and 43.29% of the sequences in the topsoil and the subsoil, respectively. Basidiomycota (41.8% and 37.5%, respectively) were also present in both soil layers. In the HN plots, Ascomycota (50.09% and 39.59%, respectively) and Basidiomycota (37.5% and 51.65%, respectively) were the most abundant phyla in the topsoil and the subsoil (Fig 3).

The relative abundance of the bacterial phyla Firmicutes and Chloroflexi increased with an increase in N addition, especially in the HN plots. At the class level, six groups varied with N addition. These include Subgroup 2 and Acidobacteria from phylum Acidobacteria, Ktedono-bacteria and JG37-AG-4 from phylum Chloroflexi, Clostridia from phylum Firmicutes, and Acidimicrobiia from phylum Actinobacteria (S3A Fig).

Two fungal families exhibited changes in abundance levels in both soil layers after N addition. One was the family Myxotrichaceae, from phylum Ascomycota, and the other was unidentified. At the genus level, the relative abundance of Oidiodendron and Cortinarius, belonging to the Ascomycota and Basidiomycota phyla, respectively, were significantly decreased in both soil layers of HN plots (S3B Fig).

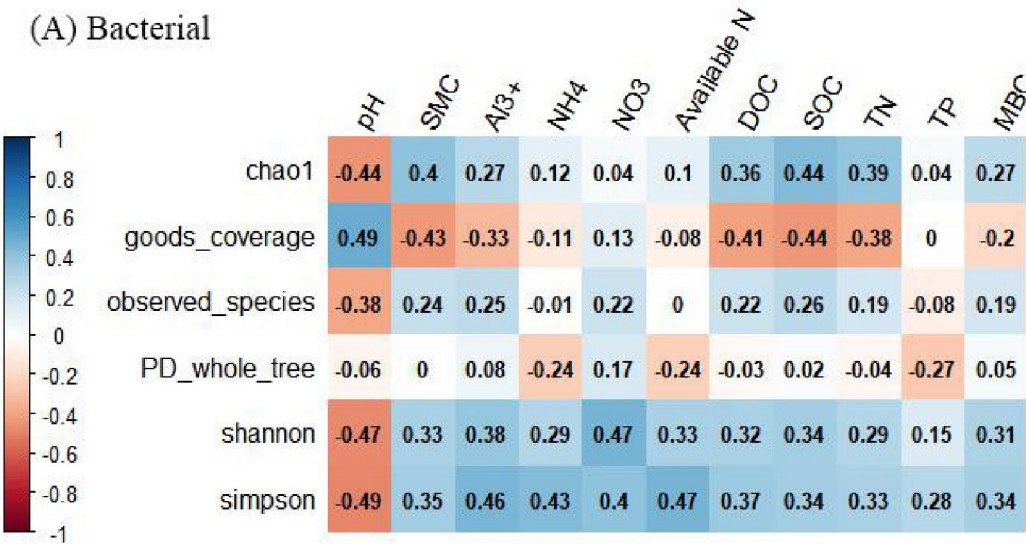

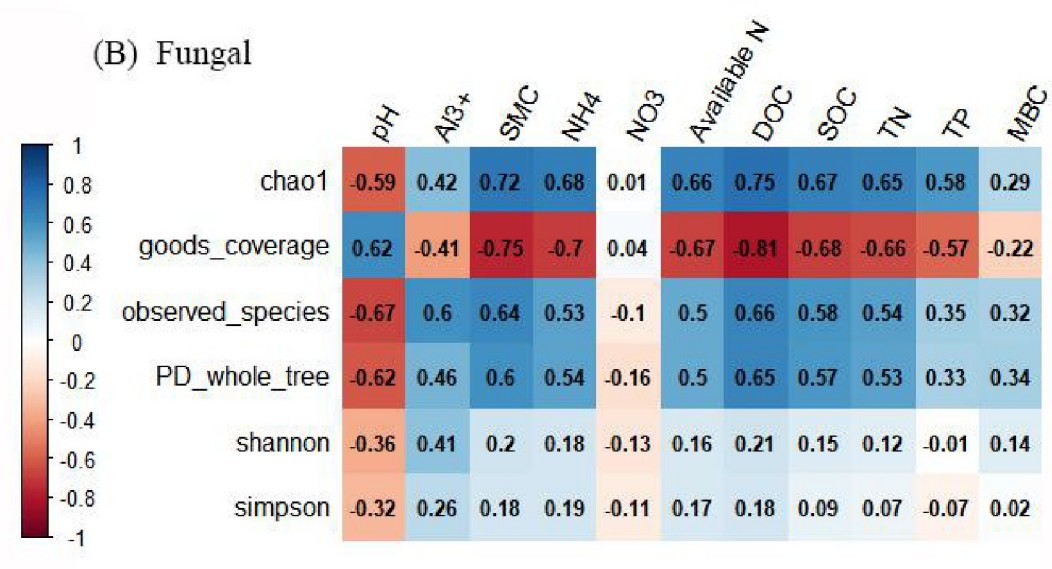

**Fig 2. Pearson's correlation coefficients between soil properties and soil bacterial (A) and fungal (B) α-diversity.** α-diversity indices include Chao1, good coverage, observed species, PD whole tree, Shannon, and Simpson indices; SMC represents soil moisture content, CEC represents cation exchange capacity, SOC represents soil organic carbon, TN represents total nitrogen, TP represents total phosphorus, CN ratio represents the carbon:nitrogen ratio, and DOC represents dissolved organic carbon.

## Bacterial and fungal community structure and correlations with environmental parameters

The results of two-dimensional RDA carried out at the OTU level (Fig 4) indicated that soil DOC was the most important parameter influencing soil bacterial community composition, and their relationship was closely correlated with the first RDA axis (Fig 4A).

RDA also revealed that soil AN and MBN from the first two constrained axes explained 36.92% of the variation in the fungal community, with the first and second axes explaining

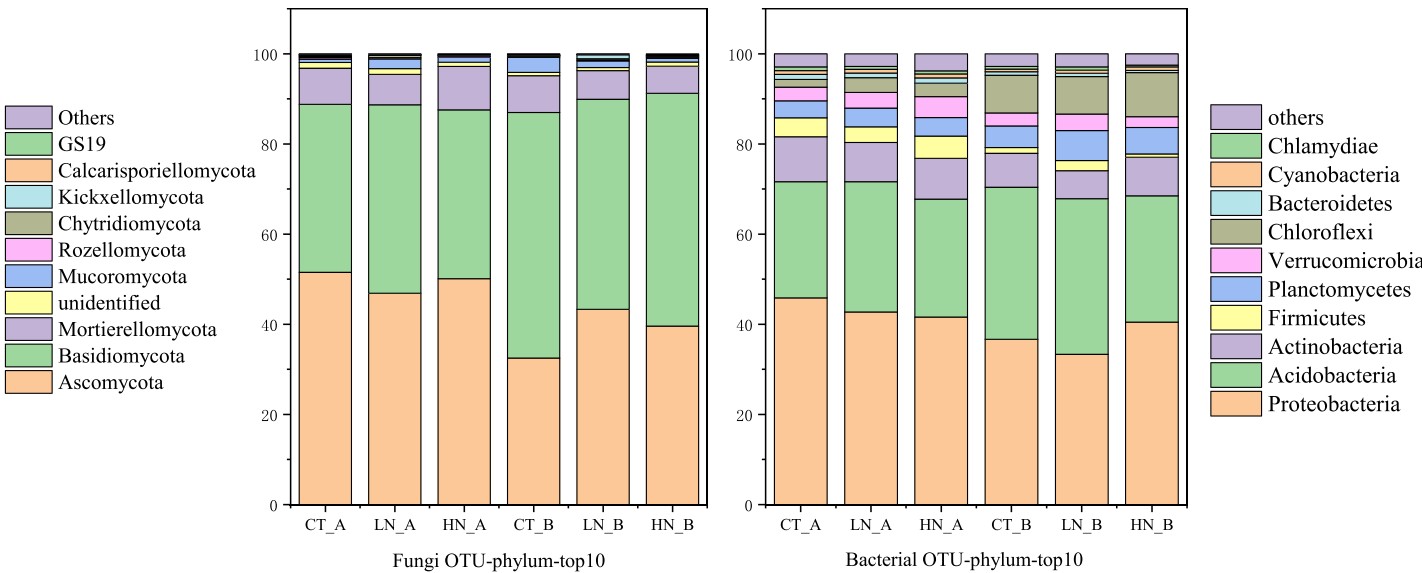

**Fig 3. Relative abundance of dominant phyla of bacteria and fungi in the topsoil and subsoil under different N addition treatments.** Control treatment in topsoil is represented by CT_A, low nitrogen addition in topsoil is represented by LN_A, high nitrogen addition in topsoil is represented by HN_A; control treatment in subsoil is represented by CT_B, low nitrogen addition in subsoil is represented by LN_B, and high nitrogen addition in subsoil is represented by HN_B.

24.54% and 12.38% of the variation, respectively. Moreover, RDA clearly showed that the soil AN and MBN concentrations were the most significant contributors to the variation in fungal communities (p < 0.01).

The Mantel test analysis of soil physicochemical properties and microbial community structure revealed that soil bacterial community structure was not significantly related to most parameters of the topsoil and the subsoil. However, bacterial community structure was significantly correlated with SMC in the subsoil (p < 0.01), and fungal community structure was significantly correlated with MBN as well as AN in the topsoil, which were consistent with the RDA results (Table 3).

## Discussion

### Effect of nitrogen addition on bacterial and fungal gene abundance and diversity

Previous studies have showed that N enrichment has positive [55, 56], neutral [47, 57], or negative [58, 59] impacts on soil microbial biomass (MBC/MBN) in tropical or subtropical forest ecosystems. Several studies have also reported that the F/B ratio increases in tropical or subtropical forest ecosystems under short-term N enrichment [33, 59, 60]. However, a recent meta-analysis illustrated that N enrichment enhanced MBC and reduced F/B in tropical or subtropical forests [28]. This suggests that there is no overall consensus regarding the effects of N addition on soil microbial diversity and biomass. In the present study, MBN was significantly increased by N addition (Table 1; p < 0.01). As soil microbes are mainly C limited [61], an increase in labile C input is expected to multiply microbial biomass. Hence, owing to CN coupling, the increase in MBN in our study is consistent with such expectations.

Previous studies have also demonstrated that N addition decreases microbial abundance and diversity [5, 24, 27, 38, 62]. In contrast, bacterial and fungal abundances in the present study were increased in the LN plots, although the increases were not significant. However, bacterial abundance estimated based on copy numbers of 16S rDNA and fungal abundance

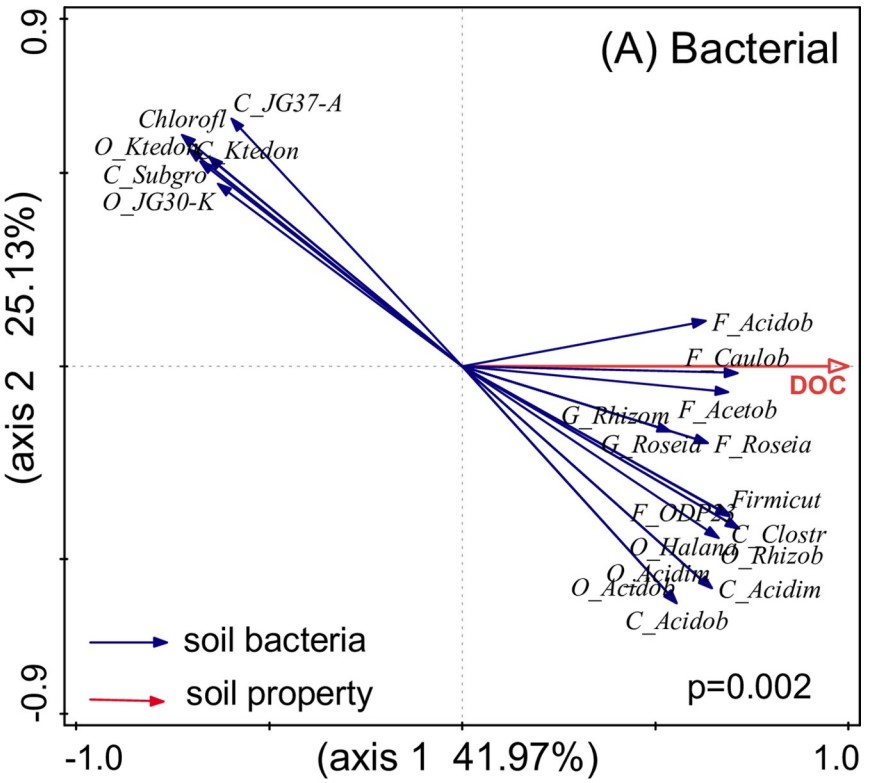

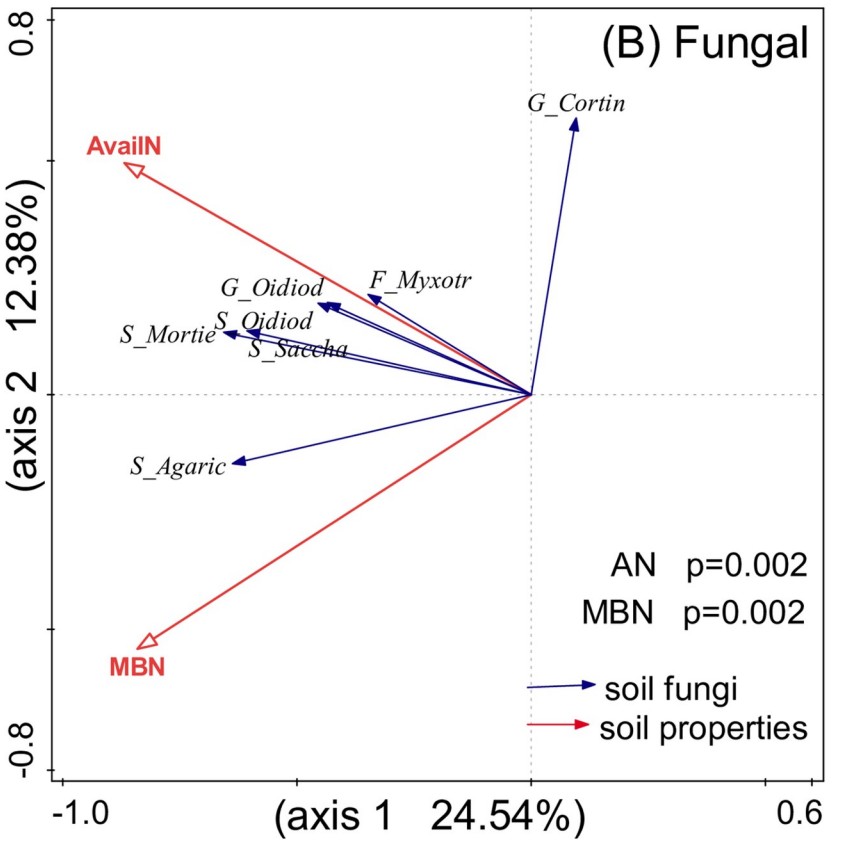

**Fig 4. Redundancy discriminate analysis (RDA) plots illustrating the relationships between the dominant bacterial (A) and fungal (B) phyla and soil physicochemical properties and microbial biomass.**

estimated based on ITS gene copy numbers were reduced in the HN plots, which is consistent with the findings of recent studies [27, 63, 64]. Pearson's correlation analysis revealed that the abundance of bacterial 16S rDNA gene and the F/B ratio were strongly correlated with soil pH in the present study, suggesting that soil pH may be a key factor influencing microbial diversity (S3 Table). Bacterial abundance was considerable influenced by edaphic pH, potentially due to the low tolerance of most bacterial groups to variation in pH (mostly within 4–7 pH) [27, 31, 40, 65]. Soil parameters that can be drastically affected by C and N supply to the soil play a vital role in determining soil bacterial diversity in global terrestrial ecosystems [66, 67]. Previous studies have shown that soil pH is a major factor influencing microbial diversity in ecosystems [35, 42, 68, 69]. However, in the present study, DOC exerted the strongest effect on soil bacterial and fungal diversity (Fig 2), and AN significantly affected soil fungal diversity. Additionally, bacterial 16S rDNA gene abundance was closely related to AN, suggesting that N availability is a major factor regulating the abundance of soil bacteria [5, 70, 71].

## Influence of nitrogen addition on bacterial and fungal community structure

In the present study, high-throughput sequencing analysis revealed that Acidobacteria, Proteobacteria, Actinobacteria, Firmicutes, and Chloroflexi are the dominant bacterial phyla present in the soils of tropical/subtropical forests, consistent with results from previous studies [5, 72]. The relative abundance of Proteobacteria and Actinobacteria decreased in the topsoil of the LN and HN plots compared to in the controls, while that of Acidobacteria and Chloroflexi increased upon N addition. These findings are not consistent with the copiotrophic hypothesis

**Table 3. Mantel test analysis of soil physicochemical properties and soil microbial biomass and community composition in topsoil and subsoil under different nitrogen addition treatments.**

| environment variable | Topsoil (0–10 cm) | | | | Subsoil (10–20 cm) | | | |
|---|---|---|---|---|---|---|---|---|
| | Bacterial community composition | | Fungal community composition | | Bacterial community composition | | Fungal community composition | |
| | r | p | r | p | r | p | r | p |
| pH | 0.005 | 0.378 | 0.162 | 0.165 | -0.186 | 0.930 | -0.063 | 0.700 |
| SMC | -0.046 | 0.475 | 0.295 | 0.112 | **0.4382** | **0.010**\*\* | 0.101 | 0.256 |
| NH$_4^+$-N | -0.137 | 0.784 | 0.253 | 0.093 | 0.016 | 0.424 | 0.127 | 0.238 |
| NO$_3^-$-N | 0.004 | 0.448 | 0.202 | 0.139 | 0.217 | 0.136 | -0.136 | 0.739 |
| AN | -0.048 | 0.543 | **0.307** | **0.041**\* | 0.145 | 0.206 | 0.105 | 0.301 |
| DOC | -0.155 | 0.738 | 0.207 | 0.130 | 0.1401 | 0.253 | -0.130 | 0.684 |
| SOC | 0.016 | 0.380 | 0.295 | 0.096 | 0.201 | 0.130 | -0.122 | 0.761 |
| TN | 0.033 | 0.351 | 0.383 | 0.080 | 0.222 | 0.160 | -0.129 | 0.722 |
| TP | -0.187 | 0.853 | -0.076 | 0.572 | 0.112 | 0.250 | -0.149 | 0.736 |
| MBC | -0.013 | 0.449 | 0.070 | 0.299 | -0.148 | 0.821 | -0.207 | 0.913 |
| MBN | -0.142 | 0.853 | **0.306** | **0.018**\* | -0.206 | 0.878 | **0.349** | **0.049**\* |

SMC, soil moisture content; CEC, soil cation exchange capacity; NH4$^+$-N, ammonium N; NO3$^-$N, nitrate N; AN, available N; DOC, dissolved organic carbon; SOC, soil organic carbon; TN, total nitrogen; TP, total phosphorus; MBC, microbial biomass carbon; MBN, microbial biomass nitrogen.

\*p < 0.05

\*\*p < 0.01.

as advocated by Fierer et al. [71]. However, the relative abundance of Proteobacteria and Actinobacteria increased in the subsoil of HN plots compared to those of the controls, while that of Acidobacteria and Firmicutes decreased upon N addition. Our results with regard to the relative abundance of different phyla in the subsoil (10–20 cm) are consistent with the copiotrophic hypothesis. Moreover, no apparent divergence was observed in the relative abundance of Proteobacteria, Acidobacteria, Chloroflexi, Actinobacteria, Verrucomicrobia, Firmicutes, and Planctomycetes between the LN and HN plots. This may be explained by the fact that soil pH remained at around 4.22–4.35 in the topsoil (Table 1), and most microbes are inhibited when pH is below 4.5 [73–75]. Since short-term N addition did not cause significant soil acidification in our experiment, we may conclude that pH was not the major factor regulating soil microbial community structure in the present study.

Actinobacteria and Proteobacteria are known to display more rapid growth rates under high C availability conditions, while Acidobacteria are oligotrophic bacteria that exist in nutrient-deficient and strongly acidic environments, and are capable of degrading recalcitrant and complex C compounds [76]. Zhang et al. [66] reported high Acidobacteria abundance under very low SOC conditions (2.66 g kg$^{-1}$). In addition, Acidobacteria have been reported to exhibit negative correlations with soil C availability [37]. The Ktedonobacteria is a distinctive class of prokaryotes that exhibits morphology similar to actinomycetes, and is thought participate in C cycling. The Ktedonobacteria class of bacteria was first reported by Cavaletti et al. [77], and classified within the phylum Chloroflexi [78], which is a diverse group of bacteria.

The class Acidimicrobiia belongs to the phylum Actinobacteria, which plays a pivotal role in the soil nutrient cycle by generating extracellular enzymes and forming symbiotic interactions with plants [79–81]. These extracellular enzymes can decompose plant litter, thus regulating C availability in the soil [82]. Several bacterial groups isolated from Actinobacteria also have a capacity to fix N and remove P from the soil [82–84]. Fungal communities are less affected by N addition than bacterial communities are, potentially because bacteria exhibit more copiotrophic characteristics [15, 29]. Soil fungal communities are a functionally diverse groups [85] that mediate numerous ecological processes and influence plant growth and soil health [86, 87]. ITS sequencing analysis results indicated that Ascomycota and Basidiomycota were the dominant fungal phyla in the acidic forest soil, supporting the copiotrophic hypothesis. Since soil nutrients decrease with an increase in soil depth, the relative abundance of Basidiomycota, which is the representative phylum of oligotrophic taxa, was lower than that of Ascomycota in the topsoil (Fig 3). Previous studies have shown that N addition enhances the relative abundance of copiotrophic phyla such as Ascomycota and reduces that of oligotrophic phyla such as Basidiomycota, which is consistent with the prediction of the copiotrophic hypothesis [34, 88–90]. The family Myxotrichaceae and the genus *Oidiodendron* participate in the decomposition of cellulose [91]. Although no significant impact of N addition on the soil fungal community structure was observed (Fig 3), RDA and Mantel test analysis results revealed that soil fungal community composition was related to AN and MBN in the soil (Fig 4, Table 3). The results suggest that N addition potential affects species diversity directly via increase of N availability.

Other physicochemical properties may also influence soil microbial community structure. For example, RDA analysis results suggested that soil bacterial community composition was significantly related to DOC concentrations, but not significantly correlated with soil pH in the subtropical acidic forests (Fig 4A). DOC is the most preferred nutrient by the vast majority of bacteria. Urea is the primary mode of N application, and its hydrolysis can consume H$^{+}$ and increase pH. NH$_4^{+}$ is often formed after urea is applied to the soil, which counteracts the H$^{+}$ enrichment caused by NO$_3^{-}$ leaching; this may why soil pH remained unchanged after short-term N application in the present study. N addition could also influence soil microbe structure

indirectly by altering soil C availability, C:N, and soil pH [19, 92]. When N addition accelerates C consumption, C supply becomes a limiting factor and the rate of lignin decomposition reduces [93]. This leads to a decline in C storage in the soil, which facilitates the metabolic activities of other heterotrophic microorganisms [94]. According to the concept of resource allocation, adequate N supply can activate C-related microbial growth, resulting in the rapid putrefaction of plant cellulose-rich litter [30, 95, 96]. The N saturation hypothesis proposes that N addition may diminish the microbial demand for additional N in terrestrial ecosystems, resulting in C or P limitation for soil microorganisms [97–99]. RDA and Mantel test analysis results showed that soil fungal community composition was correlated with soil AN and MBN ($p < 0.01$) (Fig 4B, Table 3), probably due to the dominant role of fungi in the rapid mineralization of N as well as microbial retention of available N [100, 101].

In the present study, PLS-DA results demonstrated that fungi and bacteria inhabited disparate ecological niches and were partitioned across different treatments (S2 Fig). We believe that shifts in available C and N, rather than the alteration of soil pH, caused the changes in microbial diversity in response to N addition. The finding is consistent with those of recent studies showing that nutrient availability is mainly responsible for the responses of the soil microbial community structure to N addition [24, 25, 102].

## Conclusion

Overall, in the present study, we showed that the composition, and diversity of soil bacterial and fungal communities were weakly influenced by short-term N addition. High-N addition reduced soil bacterial diversity in the subsoil and increased the relative abundance of oligotrophic bacteria in the soils of extremely acidic subtropical forests. Notably, N addition had positive impacts on some bacterial groups (e.g., Ktedonobacteria and Acidobacteria) involved in C cycling. Soil microbial community composition was also associated with soil physicochemical properties such as DOC, AN, $NH_4^+$, and $NO_3^-$. N addition influence soil microbial community structure mainly by increasing nutrient availability rather than via edaphic acidification. Soil microbes are critical components of biogeochemical cycles, and possess the capacity to transform soil nutrients (non-available N and C) into usable forms (available N and C). Microbial community responses to N addition are long-lived and gradual processes, which may vary over time. Hence, the effects of chronic N addition on microbial communities and their interactions with biogeochemical cycles in such subtropical forest ecosystems should be examined further. Moreover, studies on the effects of N addition on biogeochemical cycles would also greatly benefit from linking such functions with microbial community structure.

## Supporting information

**S1 Fig.** Rarefaction curves of bacterial (A) and fungal (B) α-diversity. The amount of sequencing data is deemed to be reasonable when the curves are flat and more data will produce only a small number of new species.
(ZIP)

**S2 Fig.** Partial Least Squares Discrimination Analysis (PLS-DA) of the soil bacterial (A, B) and fungal (C, D) community structure in the topsoil and subsoil under different N addition treatments. The variation in community composition was determined based on the abundance of OTUs. CT, LN, and HN, represent the control, low, and high nitrogen addition treatments, respectively.
(ZIP)

**S3 Fig.** Cladograms depicting bacterial (A) and fungal (B) species variation under different N addition treatments. LDA analysis was performed to obtain these cladograms.
(TIF)

**S1 Table. Primer sequences used in this study.**
(DOCX)

**S2 Table. Effects of different N addition treatments at different soil depths based on 16S rDNA gene copy numbers, ITS gene copy numbers, and fungi-to-bacteria ratios.**
(DOCX)

**S3 Table. Pearson's correlations of microbial abundance with soil properties and microbial biomass.**
(DOCX)

**S4 Table. Good's coverage values of each group.**
(DOCX)

**S5 Table. Effects of N addition on soil bacterial and fungal α-diversity indices.**
(DOCX)

## Acknowledgments

We thank Prof. Zhigao Sun and Prof. Quanlin Zhong for encouraging the development of this study and to Yating Chen and Liangtai Zhou for their help in field work.

## Author Contributions

**Conceptualization:** Juyan Cui, Xiaochun Yuan, Kaimiao Lin, Jianguo Xu.

**Formal analysis:** Juyan Cui, Xiaochun Yuan.

**Investigation:** Juyan Cui, Xiaochun Yuan, Yue Wu, Lei Cheng, Quanxin Zeng, Kongcan Mei.

**Methodology:** Juyan Cui, Qiufang Zhang, Jiacong Zhou, Kaimiao Lin.

**Software:** Jiacong Zhou, Yaozhong Zeng.

**Visualization:** Juyan Cui, Yaozhong Zeng.

**Writing – original draft:** Juyan Cui.

**Writing – review & editing:** Qiufang Zhang, Yuehmin Chen.

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
