## [Decision Letter · Decision Letter 0]

17 Sep 2020

PONE-D-20-10592

Nutrient availability is a dominant predictor of soil bacterial and fungal community composition after nitrogen addition in subtropic al acidic forest

PLOS ONE

Dear Dr. Chen,

Thank you for submitting your manuscript to PLOS ONE. After careful consideration, we feel that it has merit but does not fully meet PLOS ONE’s publication criteria as it currently stands. Therefore, we invite you to submit a revised version of the manuscript that addresses the points raised during the review process.

As noted by all reviewers the manuscript requires major revisions, especially with respect to the written language, and the presentation and discussion of the study. The authors should address all reviewers' comments in a revised manuscript..

We look forward to receiving your revised manuscript.

Kind regards,

Julian Aherne

Academic Editor

PLOS ONE

Additional Editor Comments:

As noted by all reviewers the manuscript requires major revisions, especially with respect to the written language, and the presentation and discussion of the study. The authors should address all reviewers' comments in a revised manuscript.

Journal Requirements:

2. Thank you for submitting the above manuscript to PLOS ONE. During our internal evaluation of the manuscript, we found significant text overlap between your submission and the following previously published works:

https://peerj.com/articles/7631/

https://www.sciencedirect.com/science/article/abs/pii/S0048969717335684?via%3Dihub

Please revise the manuscript to rephrase the duplicated text, cite your sources, and provide details as to how the current manuscript advances on previous work. Please note that further consideration is dependent on the submission of a manuscript that addresses these concerns about the overlap in text with published work.

4. In your Methods section, please provide additional location information, including geographic coordinates for the data set if available.

5. We note that Figure 1 in your submission contain map images which may be copyrighted. All PLOS content is published under the Creative Commons Attribution License (CC BY 4.0), which means that the manuscript, images, and Supporting Information files will be freely available online, and any third party is permitted to access, download, copy, distribute, and use these materials in any way, even commercially, with proper attribution. For these reasons, we cannot publish previously copyrighted maps or satellite images created using proprietary data, such as Google software (Google Maps, Street View, and Earth). For more information, see our copyright guidelines: http://journals.plos.org/plosone/s/licenses-and-copyright.

5.1. You may seek permission from the original copyright holder of Figure 1 to publish the content specifically under the CC BY 4.0 license. 

5.2. If you are unable to obtain permission from the original copyright holder to publish these figures under the CC BY 4.0 license or if the copyright holder’s requirements are incompatible with the CC BY 4.0 license, please either i) remove the figure or ii) supply a replacement figure that complies with the CC BY 4.0 license. Please check copyright information on all replacement figures and update the figure caption with source information. If applicable, please specify in the figure caption text when a figure is similar but not identical to the original image and is therefore for illustrative purposes only.

6. We note that you are reporting an analysis of a microarray, next-generation sequencing, or deep sequencing data set. PLOS requires that authors comply with field-specific standards for preparation, recording, and deposition of data in repositories appropriate to their field. Please upload these data to a stable, public repository (such as ArrayExpress, Gene Expression Omnibus (GEO), DNA Data Bank of Japan (DDBJ), NCBI GenBank, NCBI Sequence Read Archive, or EMBL Nucleotide Sequence Database (ENA)). In your revised cover letter, please provide the relevant accession numbers that may be used to access these data. For a full list of recommended repositories, see http://journals.plos.org/plosone/s/data-availability#loc-omics or http://journals.plos.org/plosone/s/data-availability#loc-sequencing.

7. Please upload a new copy of Figure 1 as the detail is not clear. Please follow the link for more information: https://blogs.plos.org/plos/2019/06/looking-good-tips-for-creating-your-plos-figures-graphics/

Reviewers' comments:

Reviewer's Responses to Questions

**Comments to the Author**

1. Is the manuscript technically sound, and do the data support the conclusions?

Reviewer #1: Yes

Reviewer #2: Yes

Reviewer #3: Partly

2. Has the statistical analysis been performed appropriately and rigorously? 

Reviewer #1: Yes

Reviewer #2: Yes

Reviewer #3: I Don't Know

3. Have the authors made all data underlying the findings in their manuscript fully available?

Reviewer #1: No

Reviewer #2: Yes

Reviewer #3: Yes

4. Is the manuscript presented in an intelligible fashion and written in standard English?

Reviewer #1: No

Reviewer #2: No

Reviewer #3: No

5. Review Comments to the Author

Reviewer #1: This study compared soil microbial community in forest soil under different N treatments. The authors found N addition only decreased bacteria α-diversity in subsoil (Note: contradictory statement L37 & L255) and soil nutrients rather than pH plays a larger role in determining community composition. While I think the data could be useful, the manuscript in its current form needs some revision before acceptation. I hope the suggestions below help the authors improve the manuscript.

-The introduction of the manuscript is well written. But I found some recent published long-term studies on N fertilization in tropical/subtropical forests were not included. Particularly, a paragraph focusing on microbial community shift caused by N addition in subtropical forests could be added. I attach a few references and hope they can be incorporated into introduction/discussion.

Wang et al., (2018) https://doi.org/10.1016/j.soilbio.2018.08.022

Wang et al., (2018) https://doi.org/10.1016/j.soilbio.2018.03.009

Wu et al., (2019) https://doi.org/10.1016/j.apsoil.2019.05.014

-Methods did not describe measurement of CEC.

-For discussion of diversity, an explanation of why only bacteria diversity was affected in subsoil is needed. Particularly I found attributing diversity change to pH (L334) WAS NOT reasonable as I found it contradictory that HN treatment decreased bacteria α-diversity in subsoil significantly (Table S5), but subsoil pH did not show significant difference in CK and HN plots. Instead, subsoil pH was significant different between CK and LN plots (Table 1). And in L395 authors stated pH was not changed making it more confusing.

-For discussion of community structure, does the community structure change somehow reflect observed functions? For instance, the observed decreased NH4 and increased NO3 in LN plots suggest potential high nitrification rate.

-L428-431 I found the concluding remark is quite general, a stronger take home message is preferred. Also a lot of studies have studied biogeochemical cycling functions under N addition, it would make a more significant contribution by connecting functions with community structure.

-The writing needs substantial work. I encourage the authors to thoroughly check the manuscript to avoid grammatical mistakes and incomplete sentences. I also suggest authors remake some tables and figures. It is very hard to read figure legends in its current layout (e.g., Fig 1-3, Fig S1-3). Fig.2 does not have legend and caption does not provide enough information to interpret figure. Fig.4 does not have a legend and data labels have inconsistent font, also these labels make the figure messy. In Table 1, use superscript letters to avoid incomplete post-hoc results. Also I am very concerned about the reported soil moisture content (SMC<1%), authors should double check their data.

-L787-789 correlation?

-Please include all data (prior to statistical analysis) in supplement or deposit to a public repository for review.

Reviewer #2: The paper by Cui et al. reveals soil microbial diversity and community structure under nitrogen (N) addition in a subtropical forest. The experiment is well conducted, but the present paper does not write well. "Introduction" and "Discussion" are not well setting out, and some conclusions are not suitable. Especially, the conclusion in "Abstract" that soil available N was significantly decreased under N addtion is not exact. In fact, soil available N was not decreased under high N addition according to their results. The duration of the experiment is relatively short (1 year) and the mean annual precipitation on this study site is relatively high (1800 mm). Soil N leaching may be high, and thus influenced results of soil available N based on one sampling time. I suggest the authors polish the English writing.

Abstract

Line 36, not exact.

Line 40, add a sentence about the results' indication.

Introduction

The "Introduction" is lack of continuity in logic, and does not show how scientific questions are proposed. Some information should be given in " Discussion".

Line 62-73, should be rewritten.

Material and methods

Line 107, subtropical. It is...

Line 109, "Pinus taiwanensis" should be in italic.

Line 211, the typeface is not consistent.

Results

Line 218-228, the results should show the data rather than description.

Line 223-224, the sentences should be rewritten. Is soil available N the sum of NH4+-N and NO3--N? Some values are the sum, and others are not in Table 1.

Line 229, as a main result, it is not suitable that tables and figures in this section are not present in the main text. In fact, I think the results of soil bacterial and fungal gene copy numbers can be deleted in the text. Because the gene copy numbers are determined by the PCR cycles. The values are not in situ absolute values.

Table 1, data should be shown as mean ± SE/SD.

Discussion

Line 314-320, what indications of these pevious studies for this study?

Line 325-331, the "Discussion" should not repeat results simply.

Reviewer #3: Cui et al. investigated soil microbial responses to one-year N addition treatments and identified the dominant factors. I believe the data is sufficient and the overall structure of the manuscript is clear. However, there is a big problem with writing. My major concerns are as follows. See the detailed comments in the attached pdf.

1, I can understand what the authors have written, but there are a lot of grammatic errors in this manuscript.

2, Although I like the three questions asked at the end of the Introduction, the section has not been well developed. I understand the structure of the section. Yet sentences within a paragraph were not tightly linked or well organized. Besides, the knowledge gap was not well developed and thus the significance of the work seems less appealing to me.

3, I encourage the authors to further explore the relationships among microbial communities and environmental parameters. Right now, they only used RDA to identify important variables for overall community composition. I cannot see the direction of the effects of these variables on specific microbial groups.

4, The whole Discussion was terribly written. It was superficial and discrete most of the time. I really hope the authors could rewrite the whole section.

6. PLOS authors have the option to publish the peer review history of their article (what does this mean?). If published, this will include your full peer review and any attached files.

Reviewer #1: No

Reviewer #2: No

Reviewer #3: No

---

## [Author Response · Author response to Decision Letter 0]

26 Oct 2020

Response to Reviewer’s Comments

Editor’s Review

Response: The manuscript has been revised with for consistency with Plos One guidelines for style. We have also provided figures and supplementary information in separate files according to guidelines and placed figure legends and tables at the appropriate places within the text. The writing has been substantially improved to meet your quality standards.

2. Thank you for submitting the above manuscript to PLOS ONE. During our internal evaluation of the manuscript, we found significant text overlap between your submission and the following previously published works:

https://peerj.com/articles/7631/

https://www.sciencedirect.com/science/article/abs/pii/S0048969717335684?via%3Dihub

Please revise the manuscript to rephrase the duplicated text, cite your sources, and provide details as to how the current manuscript advances on previous work. Please note that further consideration is dependent on the submission of a manuscript that addresses these concerns about the overlap in text with published work.

Response: Thank you for your comments. We have rewritten the introduction and discussion and revised passages that may be similar to text from previous publications.

3.In your Methods section, please provide additional information regarding the permits you obtained for the work. Please ensure you have included the full name of the authority that approved the field site access and, if no permits were required, a brief statement explaining why.

Response: The corresponding author, Professor Yuemin Chen, has established a stable relationship with Daiyunshan Nature Reserve for cooperation in scientific research, and accordingly established an expert workstation within this reserve. Additionally, a member of the Administration Bureau of the Daiyunshan Nature Reserve is a co-author on this paper. Therefore, no permits were required for access to the field site. The sentence is on Page 9, Lines 145-146.

4.In your Methods section, please provide additional location information, including geographic coordinates for the data set if available.

Response: We have added accurate information about the longitude and latitude for the location on Page 8, Lines 125-127.

5. We note that Figure 1 in your submission contain map images which may be copyrighted. All PLOS content is published under the Creative Commons Attribution License (CC BY 4.0), which means that the manuscript, images, and Supporting Information files will be freely available online, and any third party is permitted to access, download, copy, distribute, and use these materials in any way, even commercially, with proper attribution. For these reasons, we cannot publish previously copyrighted maps or satellite images created using proprietary data, such as Google software (Google Maps, Street View, and Earth). For more information, see our copyright guidelines: http://journals.plos.org/plosone/s/licenses-and-copyright.

Response: Thank you for your comments. Figure 1 is drawn by Zeng Yaozhong using cartographic software (ArcGIS 10.2, Esri, USA). The map data is DEM elevation data downloaded from the geospatial data cloud of China.

Website: http://www.gscloud.cn/home#page1/4

6.We note that you are reporting an analysis of a microarray, next-generation sequencing, or deep sequencing data set. PLOS requires that authors comply with field-specific standards for preparation, recording, and deposition of data in repositories appropriate to their field. Please upload these data to a stable, public repository (such as ArrayExpress, Gene Expression Omnibus (GEO), DNA Data Bank of Japan (DDBJ), NCBI GenBank, NCBI Sequence Read Archive, or EMBL Nucleotide Sequence Database (ENA)). In your revised cover letter, please provide the relevant accession numbers that may be used to access these data. For a full list of recommended repositories, see http://journals.plos.org/plosone/s/data-availability#loc-omics or http://journals.plos.org/plosone/s/data-availability#loc-sequencing.

Response: Thank you for your comments. We have uploaded the data as requested in NCBI Sequence Read Archive. The accession number that can be used to access these SRA data is PRJNA666625 and the SRP ID is SRP286064.

7.Please upload a new copy of Figure 1 as the detail is not clear. Please follow the link for more information: https://blogs.plos.org/plos/2019/06/looking-good-tips-for-creating-your-plos-figures-graphics/

Response: Thank you for your comment. Figure 1 has been recreated to ensure that the details are clear.

Response to Reviewer’s Comments Reviewer # 1

The authors would like to thank you for the valuable suggestions. The manuscript has been revised with due attention to your comments. The writing has been substantially improved to meet your publication quality standards.

Specific Comments:

-The introduction of the manuscript is well written. But I found some recent published long-term studies on N fertilization in tropical/subtropical forests were not included. Particularly, a paragraph focusing on microbial community shift caused by N addition in subtropical forests could be added. I attach a few references and hope they can be incorporated into introduction/discussion.

Wang et al., (2018) https://doi.org/10.1016/j.soilbio.2018.08.022

Wang et al., (2018) https://doi.org/10.1016/j.soilbio.2018.03.009.

Wu et al., (2019) https://doi.org/10.1016/j.apsoil.2019.05.014

Response: We have revised the introduction according to your comments and have cited the suggested references [29-31]. We have also added a paragraph focusing on microbial community shift caused by N addition in subtropical forests. These revisions can be found on Page 5-6, Lines 74-90.

-Methods did not describe measurement of CEC.

Response: We have added details regarding the measurement of CEC in the Materials and Methods section. Please see the revised text on Page 11, lines 181-182.

-For discussion of diversity, an explanation of why only bacteria diversity was affected in subsoil is needed. Particularly I found attributing diversity change to pH (L334) WAS NOT reasonable as I found it contradictory that HN treatment decreased bacteria α-diversity in subsoil significantly (Table S5), but subsoil pH did not show significant difference in CK and HN plots. Instead, subsoil pH was significant different between CK and LN plots (Table 1). And in L395 authors stated pH was not changed making it more confusing.

Response: Thank you for the valuable comments. We have revised this paragraph as suggested. We have mentioned that DOC exerted the strongest effect on soil bacterial and fungal diversity, while pH does not appear to be as closely related to bacterial diversity. Please see the revised text on Page 25, Lines 419-420.

-For discussion of community structure, does the community structure change somehow reflect observed functions? For instance, the observed decreased NH4 and increased NO3 in LN plots suggest potential high nitrification rate.

Response: Thank you for the valuable comments. We have revised this statement as suggested and incorporated discussion regarding whether the changes in the nutrients may reflect differences in microbial community functions such as nitrification rate. Please see the revised text on Pages 28-29, Lines 478-499.

-L428-431 I found the concluding remark is quite general, a stronger take home message is preferred. Also a lot of studies have studied biogeochemical cycling functions under N addition, it would make a more significant contribution by connecting functions with community structure.

Response: We appreciate your suggestion. We have added a concluding paragraph giving a strong take-home message. Briefly, we mention that soil microbial community is affected by N addition, especially increasing oligotrophic bacteria and some microbial groups that play major roles in nutrient cycles. We have also highlighted the need for further studies connecting microbial community structure and functions and their interaction with biogeochemical cycles when subjected to N addition. Please see the revised text on Pages 29-30, Lines 508-524.

-The writing needs substantial work. I encourage the authors to thoroughly check the manuscript to avoid grammatical mistakes and incomplete sentences. I also suggest authors remake some tables and figures. It is very hard to read figure legends in its current layout (e.g., Fig 1-3, Fig S1-3). Fig.2 does not have legend and caption does not provide enough information to interpret figure. Fig.4 does not have a legend and data labels have inconsistent font, also these labels make the figure messy. In Table 1, use superscript letters to avoid incomplete post-hoc results. Also I am very concerned about the reported soil moisture content (SMC<1%), authors should double check their data.

Response: Thank you for the valuable comments. We have remade some tables and figures and added detailed figure legends and data labels. We have also attempted to make figure 4 less messy and maintain consistency in font. We have also used superscript letters for depicting post-hoc results in Table 1. Please see the revised figures 1-4 and tables 1-3. Regarding the soil moisture content, we inaccurately reported proportions instead of percentages and have now rectified this error.

-L787-789 correlation?

Response: We have revised this statement as suggested by the reviewer. Please see the revised text on Page 18, Lines 300 and refer to Tables 2 and S3 Table.

-Please include all data (prior to statistical analysis) in supplement or deposit to a public repository for review.

Response: We have uploaded the data as requested in the NCBI Sequence Read Archive. Accession number of this SRA data is PRJNA666625 and SRP ID is SRP286064.

Response to Reviewer’s Comments Reviewer # 2

We thank the reviewer for the useful comments. We have extensively revised the manuscript according to your suggestions. We hope you will find the revised manuscript to be suitable for publication in Plos one.

Abstract

Line 36, not exact.

Response: Thank you for pointing this out. We have rewritten this sentence to clearly mention that soil dissolved organic C (DOC) and available N (AN) were significantly decreased after N additions in both topsoil and subsoil. Please see the revised text on Page 3, Lines 42-43.

Line 40, add a sentence about the results' indication.

Response: Thank you for your suggestion. We have added the sentence “This experiment provides further evidence in support of shifts in the diversity and structure of soil microbial communities in acidic subtropical forests in response to increasing nitrogen deposition.” Please see the revised text on Page 4, Lines 49-51.

Introduction

The "Introduction" is lack of continuity in logic, and does not show how scientific questions are proposed. Some information should be given in " Discussion". Line 62-73, should be rewritten.

Response: The authors would like to thank the reviewer for the valuable comments. The paragraph mentioned by the reviewer has been rewritten. We have added a paragraph focusing on microbial community shift caused by N addition in subtropical forests. Subsequently, we have mentioned the scientific questions and the accompanying hypotheses that we attempted to address. We hope that the revised version of the manuscript has better logical continuity.

Material and methods

Line 107, subtropical. It is...

Response: We have revised this statement as suggested. Please see the revised text on Page 8, Lines 128. 

Line 109, "Pinus taiwanensis" should be in italic.

Response: Thank you for your advice. We have italicized all instances where this species is mentioned as suggested. Please see the revised text on Page 8, Lines 130-133. 

Line 211, the typeface is not consistent.

Response: Thank you for identifying this inconsistency. We have ensured consistency in the font and spacing throughout the manuscript. 

Results

Line 218-228, the results should show the data rather than description.

Response: Thank you for your comment. We have modified the results section accordingly to include data. Please see the revised text on Page 14, Lines 258-261. 

Line 223-224, the sentences should be rewritten. Is soil available N the sum of NH4+-N and NO3--N? Some values are the sum, and others are not in Table 1.

Response: We have revised this statement as suggested. Available N is the sum of ammonium and nitrate and we have verified the values of these parameters. Please see the revised text on Page 10, 14 Lines 176-177, 262 and the new Table 1. 

Line 229, as a main result, it is not suitable that tables and figures in this section are not present in the main text. In fact, I think the results of soil bacterial and fungal gene copy numbers can be deleted in the text. Because the gene copy numbers are determined by the PCR cycles. The values are not in situ absolute values.

Response: This paragraph has been deleted in the text according to the reviewer’s comment. We have now used copy numbers only to provide an estimate of relative abundance of different species. Please see the revised text on Page 13, Lines 262-263. 

Table 1, data should be shown as mean ± SE/SD.

Response: We have revised the data as suggested. Please see the new Table 1. 

Discussion

Line 314-320, what indications of these previous studies for this study?

Response: These previous studies indicated that there is no consensus on the response of soil microbial biomass to nitrogen addition in subtropical and tropical forest ecosystems. We have added such a statement in the discussion section. Please see the revised text on Page 24, Lines 399-400.

Line 325-331, the "Discussion" should not repeat results simply.

Response: This descriptive sentence has been deleted from the text and the possible implications of the results have been discussed instead. Please see the revised text on Pages 24-25, Lines 410-422.

Response to Review Comments Reviewer # 3

The authors would like to thank you for the insightful comments. The manuscript has been substantially revised with careful consideration of your suggestions. I hope you will find the revised manuscript to be suitable for publication in Plos One.

1.I can understand what the authors have written, but there are a lot of grammatic errors in this manuscript.

Response: The authors would like to thank you for the valuable comments. The manuscript has been rewritten and reviewed to ensure grammatical correctness. 

2. Although I like the three questions asked at the end of the Introduction, the section has not been well developed. I understand the structure of the section. Yet sentences within a paragraph were not tightly linked or well organized. Besides, the knowledge gap was not well developed and thus the significance of the work seems less appealing to me.

Response: Thank you for your constructive comments. We have added a paragraph focusing on microbial community shifts caused by N addition in subtropical forests. Subsequently, we have presented our scientific questions and corresponding hypotheses. We hope that the organization of the text and the continuity of ideas have been improved in this version of the manuscript. I have added to the end of the introduction (beginning of last paragraph) the following sentence: “Although the effects of N addition have been studied, it is not clear if the changes in microbial community are mediated by direct increases in N availability, changes in the nutrient composition or acidification of the soil.” Please see the revised text on Page 5-6, Lines 74-90 and Page 7, Lines 105-108. 

3. I encourage the authors to further explore the relationships among microbial communities and environmental parameters. Right now, they only used RDA to identify important variables for overall community composition. I cannot see the direction of the effects of these variables on specific microbial groups.

Response: Thank you for your constructive comments. We have now used the Mantel test analysis to understand the association of the soil environmental factors with the composition of soil microbial communities. Please see the added test results on Page 22, Lines 379-384.

4.The whole Discussion was terribly written. It was superficial and discrete most of the time. I really hope the authors could rewrite the whole section.

Response: We have rewritten the entire discussion section as suggested by the reviewer. We hope there is greater depth and logical continuity in this revised version.

Minor Revisions:

1.line27: You could clarify sampling time here.

Response: We have revised this sentence to include the sampling time. Please see the revised text on Page 3, line 32.

2.Line29: It would be more clear to specify how long had the N addition lasted before sampling for microbial analyses? 

Response: We have revised this statement as suggested. We sampled the soil sampling one year after N addition. Please see the revised text on Page 3, line 39.

3.Line35: Spell the full word when it first appears.

Response: We have ensured that all abbreviations have been expanded at first mention. We have clarified that DOC refers to soil dissolved organic carbon. Please see the revised text on Page 3, line 42-43.

4.Line37-39: This result is too general. Clarify what variable is important for bacterial and fungal communities, respectively.

Response: We found that DOC is most important for bacterial communities while AN is most important for fungal communities. We have modified the statement to include these details. Please see the revised text on Page 3, lines 45-47.

5.Line50: Why is this negative?

Response: Thank you for your question. Nitrogen addition has resulted in the reduction of understory vegetation. Therefore, this value is negative. This has been clarified. Please see the revised text on Page 5, lines 62.

6.Line60: Why would you use additionally here? Do you have another reason explaining why N addition could increase MBC in subtropical forests?

Response: We have removed the word “additionally” since it was not appropriate, as pointed out by the reviewer. Please see the revised text on Page 5, line 72.

7.Line62: You should give this paragraph a topic sentence.

Response: We have added the following sentence: N addition can also induce changes in soil microbial α-diversity and microbial community composition. Please see the revised text on Page 5, line 74.

8.Line79: This sentence is not clear. Do you mean absolute abundance of major microbial groups were positively correlated with nutrient concentrations? 

Response: Yes, we have rewritten the sentence for clarity. Please see the revised text on Page 7, line 99-100.

9.Line82: This sentence should go to the beginning of the paragraph.

Response: We have moved this sentence as suggested. Please see the revised text on Page 6, line 91.

10.Line89: I am quite confused by the first two sentences. What is your point here? Besides, many studies have investigated microbial responses to nutrient addition.

Response: We have deleted these two sentences. Please see the revised text on Page 7.

11.Line97-101: I like these questions. They are well linked.

Response: Thank you for your appreciation.

12.Line108: citation?

Response: This sentence was quoted [45] from Chinese geography textbook. Author: Zhao Ji, Chen Chuankang. Press: Higher Education Press. 

13.Line119: Do these plots have similar elevations, topography/slope, and facing directions? I believe these factors all affect microbial communities. I know it may be not easy to find a big area with all the above-mentioned factors uniform on mountains. But the authors should introduce these conditions.

Response: The experimental plots are close to the top of the mountain, face the same direction, and have similar slopes and elevations. We have mentioned this in the methods section. Please see the revised text on Page 9, Lines 147-148.

14.Line135: Please clarify that you used air-dried soils

Response: We have clarified this as suggested. Please see the revised text on Page 10, Lines 165.

15.Line139:C should come first.

Response: We have revised the statement as suggested. Please see the revised text on Page 10, Lines 168-169.

16.Line143: Why would you use 'exchangeable' before ammonium and nitrate?

Response: We have deleted the word “exchangeable.” Please see the revised text on Page 10, Lines 175.

17.Line174: This paragraph is a little too simple compared the qPCR paragraph. I am not encouraging you to digging into every details, but at least we should see the key details here.

Response: We have revised this paragraph as suggested by including relevant details. Please see the revised text on Page 12, Lines 209-213.

18.Line175: Do you mean quality score per sequence? Clarify.

Response: Yes, we did mean quality score per sequence. We have clarified this. Please see the revised text on Page 12, Lines 210-211.

19.Line180: Did you rarefy the OTU table?

Response: Yes, we have done so. The rarefaction curves are presented in S1 Fig.

20.Line181: Version number?

Response: The version number of this software is 10.3. We have included this detail as suggested. Please see the revised text on Page 12, Lines 219.

21.Line184: Some repeated analyses in the first and third paragraphs are confusing me.

Response: We have revised these paragraphs and removed redundant statements as suggested. Please see the revised text on Pages 12-13, Lines 222-229 and Page 13 Lines 239-243.

22.Line192: Why PCA instead of more common ordination analyses in microbial ecology such as NMDS/RDA/CCA?

Response: PCA is a method of ordination using only species composition, and is one of the unconstrained methods of ordination. CCA and RDA are two constricted ordination methods that use both species and environmental factors to ordinate data. We have provided an explanation for using this in the methods section on Page 13, Lines 231-232.

23.Line193: Do you mean relative abundance of OTUs? If yes, clarify.

Response: Here, we used the relative abundance of OTUs to calculate species composition and abundance of the microbes. We have provided the raw data in the S2 Table.

24.Line219: Did you mention all these properties in the Materials and methods section? For example, total P and CEC.

Response: We have added the methods that we used for estimating total P and CEC in the methods section. Please see the revised text on Page 10-11, Lines 170-182.

25.Line224: Do you mean the sum of ammonium and nitrate? Please clarify. 

Response: Yes. Available N is the sum of ammonium and nitrate. Please see the revised text on Page 10, Lines 176.

26.Line243: you said negative?

Response: Yes. We have revised the sentence and added the minus sign. Please see the revised text on Page 15, Lines 271.

27.Line247: It seems like you did not rarefy sequences before calculate alpha diversity indices. Can you justify this?

Response: Thank you for your comment. The Allwegene sequencing company provided us with the following alpha diversity indices: Chao1 index, Observed species index, PD whole tree index, Shannon, Simpson index. The sequences were rarefied prior to calculating these indices. The total number of rarefaction is 18402. The rarefaction curves are presented in S1 Fig.

28.Line268: What about the associations among fungal alpha diversity indices and soil properties? You showed the results in Fig. 2B but you totally neglected these results.

Response: We have added sentences explaining these associations. The pH was negatively correlated with fungal Chao1, Observed species, and PD whole tree indices, and positively correlated with the good coverage index. Soil properties (SMC, NH4+, AN, SOC, TN, DOC) were negatively correlated with fungal diversity indices (Chao1, Observed species, PD whole tree) and positively correlated with the good coverage index. We also observed weak associations of Shannon and Simpson indices with soil properties. Please see the revised text on Page 19, Lines 308-314.

29.Line776: Fig 4. Are these environmental parameters all significant? If not, please remove the insignificant ones.

Response: Some of these parameters are statistically significant. We have removed the insignificant environmental parameters and re-drawn Figure 4. Please see the revised Figure 4.

30. Line804: Fig. S3. By lefse, you mean LDA in the Materials and methods section? Please keep consistency throughout the manuscript.

Response: Yes, we have clarified this method of analysis and consistently used LDA as suggested. Please see the revised text on Page 47, Lines 877-878.

---

## [Decision Letter · Decision Letter 1]

30 Dec 2020

PONE-D-20-10592R1

Nutrient availability is a dominant predictor of soil bacterial and fungal community composition after nitrogen addition in subtropic al acidic forest

PLOS ONE

Dear Dr. Chen,

Thank you for submitting your manuscript to PLOS ONE. After careful consideration, we feel that it has merit but does not fully meet PLOS ONE’s publication criteria as it currently stands. Therefore, we invite you to submit a revised version of the manuscript that addresses the points raised during the review process.

The revised manuscript is much improved but still requires minor revisions, specifically focusing on the three points from Reviewer #2, who notes that "The results do not strongly support their main conclusions". Further, as noted by Reviewer #1, the manuscript still requires additional language editing.

We look forward to receiving your revised manuscript.

Kind regards,

Julian Aherne

Academic Editor

PLOS ONE

Additional Editor Comments (if provided):

The manuscript requires minor revisions, specifically focusing on three points noted by Reviewer #2, who notes that "The results do not strongly support their main conclusions". Further, as noted by Reviewer #1, the manuscript still requires additional language editing.

Reviewers' comments:

Reviewer's Responses to Questions

**Comments to the Author**

1. If the authors have adequately addressed your comments raised in a previous round of review and you feel that this manuscript is now acceptable for publication, you may indicate that here to bypass the “Comments to the Author” section, enter your conflict of interest statement in the “Confidential to Editor” section, and submit your "Accept" recommendation.

Reviewer #1: (No Response)

Reviewer #2: (No Response)

Reviewer #3: All comments have been addressed

2. Is the manuscript technically sound, and do the data support the conclusions?

Reviewer #1: Yes

Reviewer #2: Partly

Reviewer #3: Yes

3. Has the statistical analysis been performed appropriately and rigorously? 

Reviewer #1: Yes

Reviewer #2: Yes

Reviewer #3: Yes

4. Have the authors made all data underlying the findings in their manuscript fully available?

Reviewer #1: Yes

Reviewer #2: Yes

Reviewer #3: Yes

5. Is the manuscript presented in an intelligible fashion and written in standard English?

Reviewer #1: No

Reviewer #2: No

Reviewer #3: Yes

6. Review Comments to the Author

Reviewer #1: Authors have substantially improved the quality of the manuscript. But the writing still has not met publication criteria, I suggest authors seek further editorial advice.

Some sentences I found need revision:

L28,31 change UNDERGROUND to BELOWGROUND

L205 MiSeq is used, but in results, authors used HiSeq, clarify.

L353,430 What statistical analysis was used to compare community structure change under N treatments.

L402-403 ” As soil microbes are mainly C limited [61], the increase of labile C input is expected to multiply microbial biomass.” The context was talking MBN, why authors bring up C? Labile C?

L477 use N loss to explain lack of response in fungi community does not make sense, it contradicts bacteria change.

L499 this sentence lacks logic and evidence. Either elaborate it, or remove it.

Fig 1. The study area should not have red shading, as the fig already has red color for elevation.

Reviewer #2: The results do not strongly support their main conclusions. 1) "DOC and AN significantly decreased after N addition to both topsoil and subsoil" is not accurate. 2) bacterial alpha-diversity in subsoil only decreased under high N addition treatment. 3) "soil DOC is the most important environmental factor for bacterial community composition while AN is the most important factor for fungal communities" are not precise. Furthermore, the r value is very small (0.1-0.3).

Reviewer #3: The authors put a lot of effort into improving the manuscript. Most of my concerns have been well addressed. I do not have further content-related comments.

7. PLOS authors have the option to publish the peer review history of their article (what does this mean?). If published, this will include your full peer review and any attached files.

Reviewer #1: No

Reviewer #2: No

Reviewer #3: No

---

## [Author Response · Author response to Decision Letter 1]

14 Jan 2021

Response to Reviewer’s Comments Reviewer # 1

The authors would like to thank you for the valuable suggestions. The manuscript has been revised with due attention to your comments. The writing has been substantially improved to meet your publication quality standards.

Minor Revisions:

-L28,31 change UNDERGROUND to BELOWGROUND

Response: Thank you for the valuable comments. We have revised this statement as suggested. Please see the revised text on Page 3, Lines 29 and 32.

-L205 MiSeq is used, but in results, authors used HiSeq, clarify.

Response: Thank you for the comment. We have revised this statement as suggested. Please see the revised text on Page 17, Lines 286 and 288.

-L353,430 What statistical analysis was used to compare community structure change under N treatments.

Response: Thank you for the valuable question. One-way analysis of variance (ANOVA) was used to determine the differences in soil microbial relative abundance, and multiple comparisons were conducted using the least significant difference (LSD) test at a significance level of p < 0.05. Please see the revised text on Page 13, Lines 270–230.

-L402-403 “ As soil microbes are mainly C limited [61], the increase of labile C input is expected to multiply microbial biomass.” The context was talking MBN, why authors bring up C? Labile C?

Response: Thank you for the valuable question. C input led to the increase of MBC, but there was no statistical significance. Owing to CN coupling, MBN increases significantly. We have revised this statement as suggested. Please see the revised text on Page 24, Lines 411–413.

-L477 use N loss to explain lack of response in fungi community does not make sense, it contradicts bacteria change.

Response: Thank you for the valuable comments. This sentence cited the other authors’ views. We did not find the available evidence to support this view; hence, the sentence has been deleted in the text. 

-L499 this sentence lacks logic and evidence. Either elaborate it, or remove it.

Response: Thank you for the valuable comments. The sentence has been deleted in the text as per the reviewer’s comment. 

-Fig 1. The study area should not have red shading, as the fig already has red color for elevation.

Response: We have modified Fig 1 as suggested by the reviewer. Please see the revised text on Page 9, Line 142.

Response to Reviewer’s Comments Reviewer # 2

We thank the reviewer for the useful comments. We have extensively revised the manuscript according to your suggestions. We hope you will find the revised manuscript to be suitable for publication in Plos One.

Specific Comments:

1) "DOC and AN significantly decreased after N addition to both topsoil and subsoil" is not accurate.

Response: Thank you for pointing this out. We have rewritten this sentence to clearly mention this and modified the result section. Please see the revised text on Page 3, 14, Lines 44–45, 265–267.

2)Bacterial alpha-diversity in subsoil only decreased under high N addition treatment. 

Response: Thank you for your suggestion. We have made the necessary changes in the abstract and result sections. Please see the revised text on Page 3, 17, 30, Lines 46–47, 299–301, 515-517.

3) Soil DOC is the most important environmental factor for bacterial community composition while AN is the most important factor for fungal communities" are not precise. Furthermore, the r value is very small (0.1-0.3).

Response: The authors would like to thank the reviewer for the valuable comments. The paragraph mentioned by the reviewer has been rewritten. We have added MBN as the predictor in fungal community shift caused by N addition in subtropical forests. We hope that the revised version of the manuscript has better logical continuity. Please see the revised text on Page 3-4, 22, Lines 47–50, 385–393.

Response to Review Comments Reviewer # 3

The authors would like to thank you for the positive comments. The manuscript has been slightly revised with careful consideration of other reviewers’ suggestions. I hope you will find the revised manuscript to be suitable for publication in Plos One.

Specific Comments:

The authors put a lot of effort into improving the manuscript. Most of my concerns have been well addressed. I do not have further content-related comments.

Response: The authors would like to thank you for the valuable and constructive comments.

---

## [Editor Report · Decision Letter 2]

18 Jan 2021

Nutrient availability is a dominant predictor of soil bacterial and fungal community composition after nitrogen addition in subtropical acidic forests

PONE-D-20-10592R2

Dear Dr. Chen,

We’re pleased to inform you that your manuscript has been judged scientifically suitable for publication and will be formally accepted for publication once it meets all outstanding technical requirements.

Kind regards,

Julian Aherne

Academic Editor

PLOS ONE

Additional Editor Comments (optional):

The authors have revised the manuscript in response to the reviewers comments. The manuscript is suitable for publication.
---

## [Editor Report · Acceptance letter]

3 Feb 2021

PONE-D-20-10592R2 

Nutrient availability is a dominant predictor of soil bacterial and fungal community composition after nitrogen addition in subtropical acidic forests 

Dear Dr. Chen:

I'm pleased to inform you that your manuscript has been deemed suitable for publication in PLOS ONE. Congratulations! Your manuscript is now with our production department. 

Kind regards, 

on behalf of

Dr. Julian Aherne 

Academic Editor

PLOS ONE